# Palladium Phthalocyanines Varying in Substituents Position for Photodynamic Inactivation of *Flavobacterium hydatis* as Sensitive and Resistant Species

Vanya Mantareva [1,*], Vesselin Kussovski [2], Petya Orozova [3], Ivan Angelov [1], Mahmut Durmuş [4] and Hristo Najdenski [2]

1 Institute of Organic Chemistry with Centre of Phytochemistry, Bulgarian Academy of Sciences, 1113 Sofia, Bulgaria; ipangelov@gmail.com
2 The Stephan Angeloff Institute of Microbiology, Bulgarian Academy of Sciences, 1113 Sofia, Bulgaria; vkussovski@gmail.com (V.K.); hnajdenski@gmail.com (H.N.)
3 National Reference Laboratory for Fish, Mollusks and Crustacean Diseases, National Diagnostic Research Veterinary Institute, 1000 Sofia, Bulgaria; nrl-fmcd@bfsa.bg
4 Department of Chemistry, Gebze Technical University, Gebze 41400, Turkey; durmus@gtu.edu.tr
* Correspondence: vanya.mantareva@orgchm.bas.bg

**Abstract:** Antimicrobial photodynamic therapy (aPDT) has been considered as a promising methodology to fight the multidrug resistance of pathogenic bacteria. The procedure involves a photoactive compound (photosensitizer), the red or near infrared spectrum for its activation, and an oxygen environment. In general, reactive oxygen species are toxic to biomolecules which feature a mechanism of photodynamic action. The present study evaluates two clinical isolates of Gram-negative *Flavobacterium hydatis* (*F. hydatis*): a multidrug resistant (R) and a sensitive (S) strain. Both occur in farmed fish, leading to the big production losses because of the inefficacy of antibiotics. Palladium phthalocyanines (PdPcs) with methylpyridiloxy groups linked peripherally (pPdPc) or non-peripherally (nPdPc) were studied with full photodynamic inactivation for 5.0 µM nPdPc toward both *F. hydatis*, R and S strains (6 log), but with a half of this value (3 log) for 5.0 µM pPdPc and only for *F. hydatis*, S. In addition to the newly synthesized PdPcs as a "positive control" was applied a well-known highly effective zinc phthalocyanine (ZnPcMe). ZnPcMe showed optimal photocytotoxicity for inactivation of both *F. hydatis* R and S. The present study is encouraging for a further development of aPDT with phthalocyanines as an alternative method to antibiotic medication to keep under control the harmful pathogens in aquacultures' farms.

**Keywords:** palladium phthalocyanines; photodynamic inactivation; multidrug resistance; gram-negative bacteria; *Flavobacterium hydatis*

## 1. Introduction

Controlling infections is a huge task in the production of aquacultures under farming conditions [1]. This form of farming has grown significantly because of the increasing global demand for sea food such as table fish, mollusks, and crustaceans [2]. However, aquaculture suffers from large production losses due to fish diseases caused by bacterial pathogens. Among the major bacterial pathogens that affect various kinds of fish are Gram-negative bacteria *Flavobacterium* sp. [3]. Factors such as water quality, the ubiquitous nature and the rapid spread of harmful pathogens, adverse environmental conditions, high stocking densities, and the different stages of the fish life cycle influence the spread of infections. These characteristics, together with the increased resistance of common pathogenic bacteria, the low activity of the known chemotherapeutic agents against such agents, and the modest number of drugs which have been licensed for fishery use make disease prevention in aquaculture a difficult task. So called flavobacteria diseases in fish are

caused by multiple bacterial species within the *Flavobacteriaceae* family, e.g.: *Flavobacterium psychrophilum*, the etiological agent of bacterial cold-water disease and rainbow trout fry syndrome [4]; *Flavobacterium columnare,* the causative agent of columnaris disease [5]; *Flavobacterium hydatis* (*F. hydatis*) species [6]; as well as other uncharacterized, yellow-pigmented bacteria [7]. In cases of acute flavobacteriosis, cumulative mortality upward of 70% can occur among infected fish stocks, and survivors may suffer poor growth and spinal abnormalities. In subacute and chronic infections, flavobacteriosis elicits lingering mortalities that can lead to continuous economic losses [8].

An effective strategy, still under intensive development, which seeks to minimize infections and to make the aquaculture industry more sustainable is based on antimicrobial photodynamic therapy (aPDT) [9–11]. Antimicrobial PDT, as a method to inactivate pathogenic bacteria, is an emerging, low-cost treatment modality for local infections. It can be applied for the treatment of infectious diseases in fish farms [10]. The method is based on the joint action of a non-toxic photosensitizing compound, light in the visible and near infrared spectrum and an oxygen medium. PDT has been documented as a potential alternative to traditional antibiotics for aquaculture because of its non-target specificity [11]. The lack of side effects and the reversal of pathogenicity, as well as the re-growth of pathogens after the procedure make PDT a method which is not susceptible to the development of resistance. The localized action leads to the cytotoxic effect being exerted only in the region of irradiation without unintended phototoxic effects on the host cells [12].

Currently, phthalocyanine compounds (Pcs) with properties suitable for antimicrobial PDT are well documented, but the method is still in the experimental stage [13,14]. The metal complexes of Pcs (MPcs) have encouraging photo physicochemical properties such as an intensive far-red absorption (>670 nm) and high triplet state quantum yields with the preferred mechanism being Type II photosensitization, i.e., the production of singlet oxygen, which has a significant impact on photocytotoxicity. Symmetrical planar Pc molecules allow a variety of structural tailoring with modifications by substitution, namely on the peripheral or non-peripheral positions, and by coordinated metal/semimetal ions [15]. Recently, two Pd(II)-phthalocyanines (PdPcs) were synthesized and characterized as suitable photosensitizers for PDT with advances over other derivatives with bivalent diamagnetic metal ions [16,17]. The regular usage of dual-light aPDT in peri-implantitis for the treatment of pathogens was recently reported in clinical practice [18]. However, intensive studies are still in progress for the development of new antimicrobial drugs involving metal containing compounds [19,20]. The need of novel photosensitizers for PDT is greater than ever, due to the increment of the resistance of pathogenic bacteria [21,22].

This study describes the efficacy of antimicrobial PDT to inactivate two new isolates of pathogenic bacteria *F. hydatis*: a multidrug-resistant (R) and a sensitive (S) strain. Both pathogenic bacteria were newly isolated from Russian sturgeon and Rainbow trout gills from the real farms. This study shows the sensitivity of these pathogens toward general groups of antibiotics. The chosen palladium phthalocyanine complexes (PdPcs) are cationic and water-soluble due to the presence of four quaternized 2-hydroxypyridine groups in the peripheral and non-peripheral positions. In addition, a zinc (II) phthalocyanine (ZnPcMe) with non-specificity as a photosensitizer toward pathogenic microorganisms is also studied for the sake of comparison.

## 2. Material and Methods

### 2.1. Phthalocyanines and Other Chemicals

Three phthalocyanine complexes of palladium or zinc (pPdPc and nPdPc, and ZnPcMe) were studied. Scheme 1 shows a summary the used synthetic procedure applied for the studied complexes [16,17]. Freshly prepared stock solutions (2 mM Pd/ZnPcs) in dimethylsulphoxide (DMSO, Uvasol) were used. The samples were stored in closed vials covered with aluminum foil. Prior the experiments, serial dilutions were created in sterile 0,05 M phosphate-buffered saline (PBS). The absorption spectra were recorded on a

Shimadzu UV–Vis 3000 (Osaka, Japan) to determine the exact concentrations of Pd- and Zn Pcs (Figure 1). All solvents and solids were purchased from chemical suppliers Fluka, Merck and Sigma-Aldrich (FOT, Sofia, Bulgaria).

**Scheme 1.** Summarized synthetic procedure of the studied Pd(II)–phthalocyanines. (i) DMF, K$_2$CO$_3$, RT, 24h; (ii) PdCl$_2$, n-pentanol, DBU, Ar, reflux, 5h and (iii) DMF, DMS, 120 °C, Ar, 3 h.

### 2.2. Bacterial Strains and Cultivation

*Flavobacterium hydatis* (*F. hydatis*) is a facultative, anaerobic, mesophilic, gram-negative bacterial pathogen that inhabits circular colonies. The strains of *F. hydatis* were newly isolated and categorized as a multidrug-resistant (R) (from sturgeon gills) and a drug-sensitive (S) from (rainbow trout gills) strain, both obtained from fish farms (Bulgaria). Both *F. hydatis* isolates were characterized based on the standard tests and mass spectrometric analyses using a MALDI-TOF apparatus (Bruker, Munich, Germany). The cultivation of *F. hydatis* was carried out on Hsu-Shots medium (tryptone, yeast extract, gelatin) and aerobically for 24 h at 28 °C on nutrient media. Cells were harvested and suspended in sterile PBS at pH 7.4 to obtain suspensions with densities of $10^9$ CFU mL$^{-1}$. Viable cells were counted in the serial dilutions on solid culture media. The number of visible colonies were counted as colonies forming units per mL (CFU. mL$^{-1}$) present on an agar plate. Values were then multiplied by the dilution factor (CFU mL$^{-1}$). The strains were grown aerobically overnight at 28 °C for 24 h before experiments. Cells were harvested by centrifugation and suspended in sterile PBS. The cell suspension was measured with absorption of 0.490

at 600 nm, which corresponded to $10^9$ CFU $mL^{-1}$. Prior to each experiment, the cell suspensions were diluted to an experimental cell density of $10^6$ CFU $mL^{-1}$.

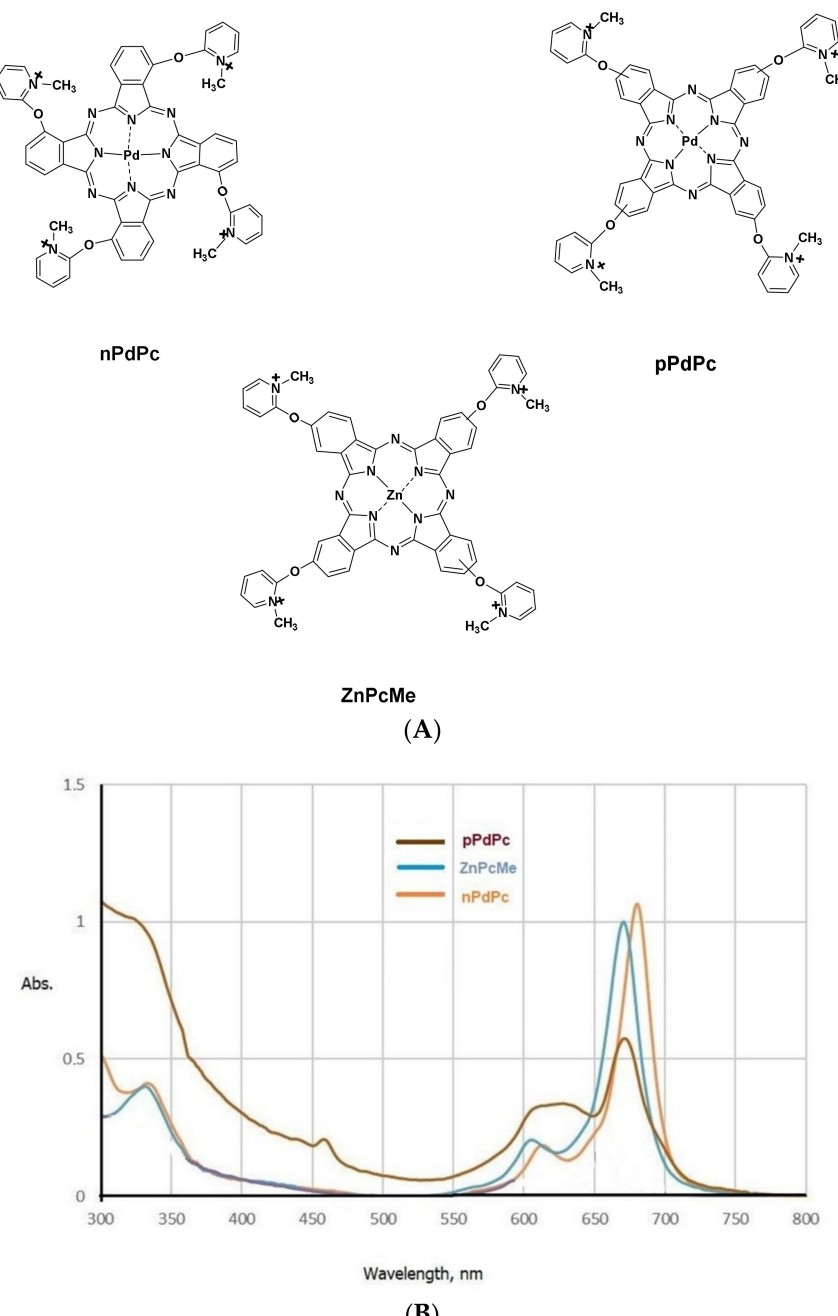

**Figure 1.** Phthalocyanine palladium (II) and zinc (II) complexes (**A**), and the absorption spectra in DMSO ($10^{-5}$ M) (**B**).

### 2.3. Antimicrobial Susceptibility

In vitro antimicrobial susceptibility was assessed using the disk diffusion method [23]. Cell suspensions of concentrations as 0.5 McFarland were prepared from cultures on R2A agar. A suspension containing each strain (100 μL) was spread on Mueller-Hinton agar plates. The tested antibiotic disks (Antimicrobial Susceptibility Disks, Oxoid) were as follows: Tobramycin (Tob$_{10}$), Novobiocin (Nb$_5$), Nalidixic acid (Nx$_{30}$), Ceftiofur (FUR $_{30}$), Enrofloxacin (EX$_5$), Gentamicin (GEN$_{10}$), Norfloxacin (NX$_{10}$), Ampicillin (AMP$_{10}$), Oxolinic acid (OA$_2$), Apramycin (APR$_{15}$), Sparfloxacin (SPX$_5$), Flumequine (FLM$_{30}$), Oxytetracycline (O$_{30}$), Co-Trimoxazole (COT$_{25}$), Florfenicol (FFC$_{30}$). CLSI (The Clinical and Laboratory

Standards Institute) and EUCAST (The European Committee on Antimicrobial Susceptibility Testing) standards were strictly followed for cultivation and inhibition zone diameter readings (CLSI, 2015; EUCAST, 2017).

*2.4. Photodynamic Inactivation Study*

Phthalocyanines with concentrations ranging from 0.04 μM to 20 μM were applied to the cell suspensions, and the cells were incubated for 15 min in a dark place. Portions of the suspension (200 μL) were placed in a standard 96-well polystyrene plate and irradiation from a light source was applied. Four groups of bacterial cells were collected as follows: (1) pure cells (no light, no Pc); (2) dark control—with Pc, without light; (3) light control—without any PS, but with light; (4) the PDI treated groups. The used light source was a light-emitting diode (LED) with wavelength maximum of 665 nm (ELO Ltd., Sofia, Bulgaria). The samples were irradiated with a fluence rate 100 mW cm$^{-2}$ for 20 min. Then, 0.1 mL was taken and diluted (10-fold) with PBS. Aliquots (0.025 mL) were spread over either Trypticase® Soy agar with 0.5% yeast extract or Hsu-Shotts agar medium. The results are presented as numbers of CFU of bacteria which developed after 48 h incubation (28 °C) on the agar dishes.

*2.5. Statistics*

All experiments with pathogenic bacteria were carried out in triplicate. The collected data are presented as mean values ± standard deviation (SD). The difference between two means was compared by an unpaired Student's test where $p < 0.05$ was considered significant.

## 3. Results

Two new clinical isolates of the pathogenic bacteria *Flavobacterium hydatis* (*F. hydatis*), i.e., multidrug-resistant (R) and sensitive (S) strains, were studied for their susceptibility to quaternized phthalocyanine complexes (nPdPc and pPdPc, and ZnPcMe, Figure 1A) designed for PDT applications. Different antibiotics were applied and the results showed the following: *F. hydatis* (R)—resistant to Tobramycin, Novobiocin, Nalidixic acid, Ceftiofur, Enrofloxacin, Gentamicin, Norfloxacin, Ampicilin, Oxolinic acid, Apramycin, Sparfloxacin, Flumequine, Oxytetracycline, Co-Trimoxazol, Florfenicol; *F. hydatis* (S)—sensitive to Nalidixic acid, Enrofloxacin, Norfloxacin, Apramycin, Sparfloxacin, Flumequine, Florfenicol; intermediate to Gentamicin, Oxolinic acid, Oxytetracycline; resitant to Tobramycin, Novobiocin, Ceftiofur, Ampicilin, Co-Trimoxazole. The antibiotic susceptibility profile of both bacterial strains used in this study is summarized in Table 1.

Two gram-negative bacterial isolates of *F. hydatis* (R and S strains) were studied for photodynamic inactivation with phthalocyanine complexes. Palladium phthalocyanine complexes (PdPcs) which differ in the position of their methylpyridiloxy groups (namely, non-peripheral for nPdPc and peripheral for pPdPc) showed an efficacy dependence on the position of the substitution groups of the ring (Figures 2 and 3). The peripheral pPdPc showed a lower photoinactivation efficiency for *F. hydatis* R strain (Figure 3). At the applied concentration range between 1 μM and 5 μM, inactivation of 3 log was observed for 2.5 μM nPdPc, while full inactivation occurred for 5 μM nPdPc for both R and S strains. In contrast, photoinactivation with 5 μM pPdPc was 2.34 log and 3.17 log against R and S strains, respectively. A light dose of irradiation of 50 J. cm$^{-2}$ and a fluence rate (power density) of 100 mW. cm$^{-2}$ were selected. These parameters showed an effective response toward *F. hydatis* R and S strains with a 6 log decrease of viable bacteria by the application of 5 μM nPdPc.

The obtained results showed a relatively high photodynamic inactivation capacity of ZnPcMe, i.e., the values of 3.12 log and 4.09 log decrease for R and S species, respectively (Figure 4). An increase of concentration to 2.5 μM and 5 μM ZnPcMe showed complete photoinactivation (>6 log decrease of viable bacteria). Dark toxicity was not observed with the applied concentration range of ZnPcMe. The results suggest that photoinactivation efficiency is not depended of the sensitivity of the *F. hydatis* strains.

**Table 1.** Antibiotic-resistant (R) and sensitive (S) strains of *Flavobacterium hydatis*.

| Antibiotics | *Flavobacterium hydatis* (Sensitive) Zone of Inhibition (mm) | *Flavobacterium hydatis* (Resistant) Zone of Inhibition (mm) |
|---|---|---|
| Tob$_{10}$ Tobramycin | 0 (R) | 0 (R) |
| Nb Novobiocin 5 µg | 0 (R) | 0 (R) |
| Nx Nalidixic acid 30 | 30 (S) | 0 (R) |
| FUR Ceftiofur 30 | 0 (R) | 0 (R) |
| EX Enrofloxacin 5 | >30 (S) | 0 (R) |
| GEN Gentamicin 10 | 12 (I) | 0 (R) |
| NX Norfloxacin 10 | 25 (S) | 0 (R) |
| AMP Ampicillin 10 | 0 (R) | 0 (R) |
| OA Oxolinic acid 2 | 15 (I) | 0 (R) |
| APR Apramycin 15 | 27 (S) | 0 (R) |
| SPX Sparfloxacin 5 | >30 (S) | 5 (R) |
| FLM Flumequine 30 | >30 (S) | 0 (R) |
| O Oxytetracycline 30 | 18 (I) | 0 (R) |
| COT Co-Trimoxazole 25 | 5 (R) | 0 (R) |
| FFC Florfenicol 30 | 28 (S) | 0 (R) |

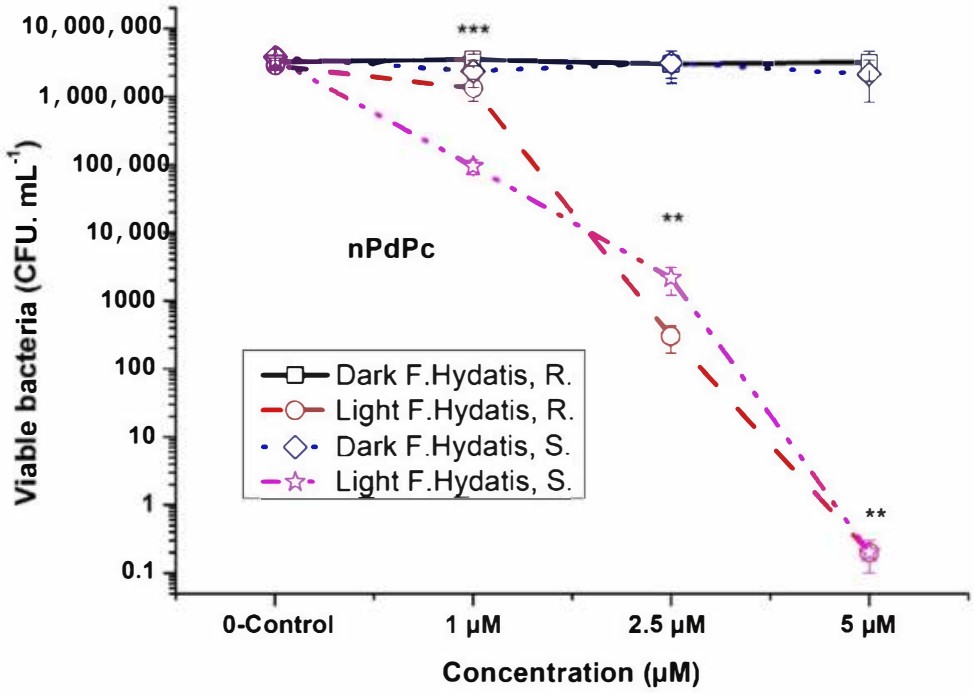

**Figure 2.** Photodynamic inactivation of *F. hydatis* (R and S) planktonic cultured with non–peripheral Pd(II)–phthalocyanine (nPdPc) and LED 665 nm irradiation. ** $p < 0.007$ and *** $p < 0.009$.

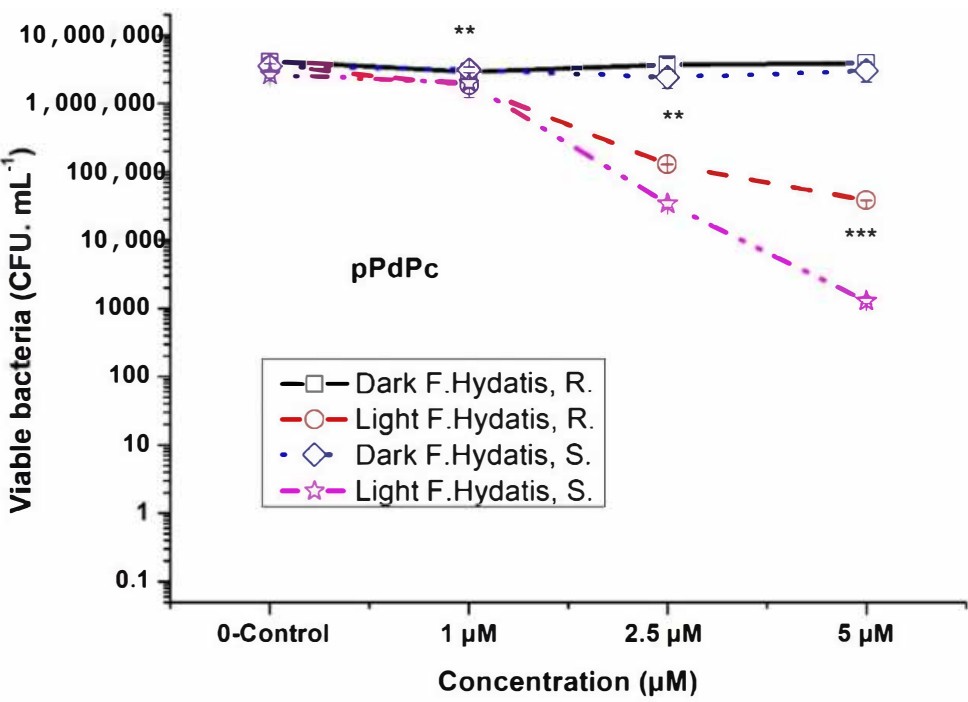

**Figure 3.** Photodynamic inactivation of *F. hydatis* (R and S) planktonic cultured with peripheral Pd(II)–phthalocyanine (pPdPc) and LED 665 nm irradiation. ** $p < 0.008$ and *** $p < 0.01$.

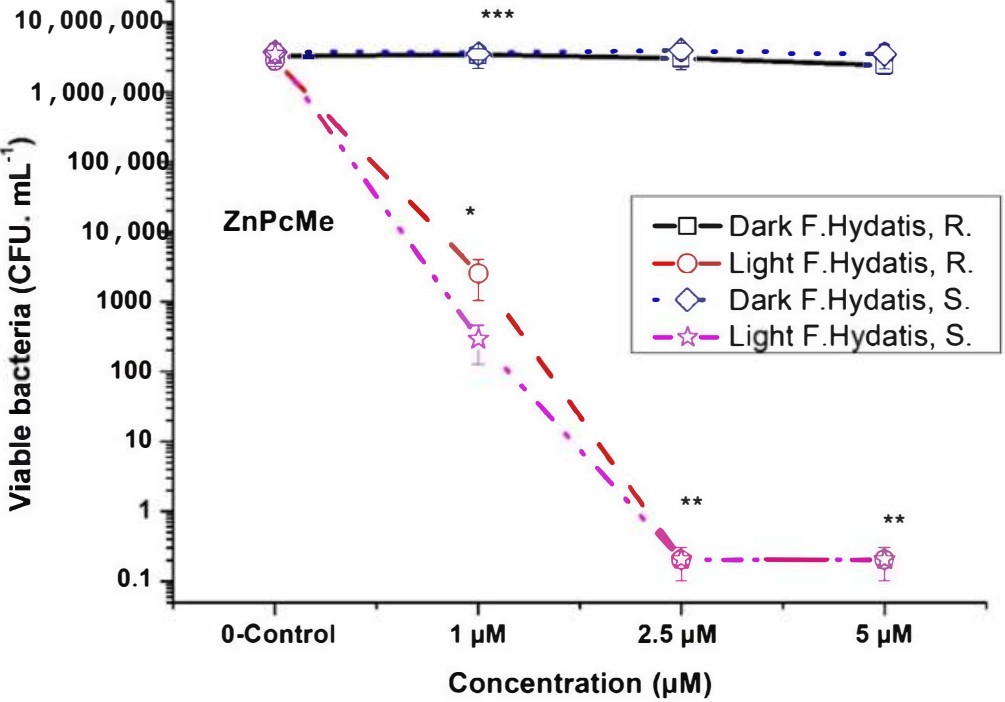

**Figure 4.** Photodynamic inactivation of *F. hydatis* (*R* and *S*) planktonic cultured with peripheral Zn(II)–phthalocyanine (ZnPcMe) and LED 665 nm irradiation. * $p < 0.005$; ** $p < 0.008$ and *** $p < 0.01$.

## 4. Discussion

Antimicrobial PDT has been considered a suitable inactivation approach to control infections in aquaculture systems [24]. This study showed that aPDT with properly selected palladium phthalocyanines inactivated sufficiently the pathogenic Gram-negative bacterial species of new isolates of *Flavobacterium hydatis* (*F. hydatis*). The high potential of aPDT was obtained for their native counterparts, also susceptible to the applied procedure.

Recent studies have suggested that pathogenic microorganisms have low ability to develop resistance to photodynamic method [25,26]. Moreover, among pathogenic bacteria the resistance did not occur even after repeated PDT exposure, and that susceptibility to antibiotics was not altered [27]. Photodynamic therapy with other photosensitizers has been found with equal efficiency in terms of controlling the growth of multiple antibiotic resistant *Vibrio harveyi* strains (responsible for luminous vibriosis, a disease that affects commercially farmed penaeid prawns) under both in vitro and in vivo conditions. For example, a procedure using Rose Bengal as the photosensitizer on an *Artemia* nauplii model (shrimp larviculture systems) and a 150 W halogen lamp with an output spectrum ranging from 450 to 600 nm was applied [10]. The study reported that PDT effectively inactivated the pathogen under both in vitro and in vivo conditions without deleterious effects on *Artemia nauplii*. A study with a cationic porphyrin (Tri-Py+-Me-PF) observed efficiency against nine strains of pathogenic bacteria (*Vibrio anguillarum*, *V. parahaemolyticus*, *Aeromonas salmonicida*, *Photobacterium damselae* subsp. *damselae*, *P. damselae* subsp. *piscicida*, *Escherichia coli*, *Pseudomonas* sp., *Enterococcus faecalis* and *Staphylococcus aureus*) isolates from a semi-intensive aquaculture system. The growth of these bacterial isolates was inhibited with a decrease of approx. 7–8 log after irradiation with visible light. The combination of porphyrin and visible light represents a promising and environmentally friendly alternative for the control of harmful pathogenic bacteria in aquaculture systems [12]. Alves et al. used the naturally luminescent marine bacterium *Vibrio fischeri* to demonstrate that a tri-cationic porphyrin could photo-inactivate *V. fischeri* under a range of different abiotic factors [12]. Arrojado et al. also showed the antimicrobial effect of a tri-cationic porphyrin on different marine bacteria, including both gram-negative and gram-positive species, in phosphate buffered saline [8]. These studies suggest the immense potential of the PDT method for application in aquaculture systems [27]. Malara et al. tested the efficacy of photolytic and photodynamic disinfection protocols using a tetra-cationic and neutral porphyrin derivatives against two pathogenic *Vibrio* species in the marine broth used for their cultivation [28]. Wong et al. studied *Vibrio vulnificus*, which frequently infects fish and contaminates fish farming waters. They showed the capacity of PDT to inactivate this pathogen [29]. Schrader et al. tested in vitro the antibacterial activity of an approved, commercially available porphyrin derivative against bacteria that are pathogenic to channel catfish, *Ictalurus punctatus* [30]. The study reported that antimicrobial PDT is also useful methodology for the photo decontamination of pools and natural lakes by inhibition of the spearing of bacterial pathogens and by preventing the growth of the resistant species [31].

## 5. Conclusions

This study establishes the further data on the efficacy of antimicrobial PDT as a very prospective alternative to antibiotics in critical circumstances with bacterial pathogens in aquacultures' farms. The obtained results indicated the susceptibility of two new clinical isolates of Gram-negative *Flavobacterium hydatis* (*F. hydatis*), i.e., a multidrug-resistant (R) strain and a drug sensitive (S) strain, to Pd(II)-phthalocyanines which differed in positions of substituents. The pathogenic species were evaluated for their sensitivity toward the main groups of antibiotics used for the prevention and treatment of acute infections which commonly occur in fish farms. Relatively high antibacterial PDT efficacy was observed with Pd(II)-phthalocyanines with methylpyridiloxy substitution groups in non-peripheral positions (nPdPc); lower efficacy was observed for phthalocyanine with substitution groups on peripheral positions (pPdPc), perhaps as a result of the long wavelength of absorption and deeper penetration of the exposed light spectrum. This study suggests that independent of the resistance of the bacterial species, the photodynamic response was similar, depending only on the photosensitizing ability of the applied phthalocyanine complex. This observation can be explained by the non-specificity of the photodynamic action. In addition, similar phthalocyanine zinc complex (ZnPcMe) was studied in comparison manner, because it is well-studied as a powerful photosensitizer for different pathogenic microorganisms. The zinc complex, ZnPcMe was shown to have higher photoinactivation efficiency towards

both bacterial isolates *F. hydatis* R and S and at lower concentrations than the applied with more effective palladium complex, nPdPc.

**Author Contributions:** Conceptualization, V.M. and V.K.; methodology, V.M. and V.K.; validation, V.K., P.O. and I.A.; formal analysis, P.O.; investigation, V.K. and P.O.; resources, V.M.; data curation, V.K. and P.O.; writing—original draft preparation, V.M. and V.K.; writing—review and editing, V.M., V.K. and M.D.; supervision, M.D. and H.N.; project administration, V.M.; funding acquisition, V.M. All authors have read and agreed to the published version of the manuscript.

**Funding:** Bulgarian National Science Fund with project KP-06-H29/11, 2018.

**Institutional Review Board Statement:** Not applicable.

**Informed Consent Statement:** Not applicable.

**Conflicts of Interest:** The authors declare no conflict of interest.

**Ethics Approval Statement:** Not applicable.

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
