# Peer review of "Palladium Phthalocyanines Varying in Substituents Position for Photodynamic Inactivation of Flavobacterium hydatis as Sensitive and Resistant Species"

_cimb, doi:10.3390/cimb44050133_

Round 1

Reviewer 1 Report

Dear authors,

This manuscript refers to biology, not chemistry. It is devoted to the isolation of two new strains of microorganisms and the proof that these new strains can be inactivated using cationic phthalocyanine complexes of palladium and zinc previously described by the authors. From the point of view of PDT methodology and chemistry, the article does not contain any new information.

The experiment was performed correctly by the authors; its description in the manuscript is quite complete. The experimental part of the work does not cause any remarks. From this point of view, I can recommend the manuscript for publication in CIMB.

Author Response

Thanks Reviewer 1 for the evaluation of our manuscript.

The novelty of this study is the isolation and characterization of the clinical species of Gram-negative bacteria as sensitive and resistant strains. The study presents also investigation on the ability of the Photodynamic method as promising approach for inactivation of the pathogenic bacterial strains. The studies suggested that PDT is not selective to the sensitivity of the stains. 

Reviewer 2 Report

The manuscript is clear and presented in a well-structured manner.

Author Response

Thanks the Reviewer 2 for the evaluation of our work and the manuscript.

Reviewer 3 Report

The antimicrobial photodynamic photodynamic therapy is a promising approach to fight the multidrug rsistance of pathogenic bacteria. The produced oxygen species are toxic to the biomolecules which features a different mechanism of action, more appropriate for the drug-resistant species. In the present study, the authors evaluates two clinical isolates of the Gram-negative F.hydatis. The authors demonstrated that the full photodynamic inactivation was observed with 5.0 μM nPdPc towards both strains F. hydatis, R and S, but a half of the inactivation value (3log) was obtained with 5.0 μM pPdPc and only for F. hydatis, S. ZnPcMe had high efficacy towards both F. hydatis R and S strains. This study gives promising base for the development of aPDT as alternative method to keep under control the farm fishes’pathogens. This is a great work and should be interested by the reader related to the International journal of Molecular Biology.

Author Response

Thank the Reviewer 3 for the positive report. Also for suggestion for another appropriate Journal.